# Highly Sensitive and Durable Structured Fibre Sensors for Low-Pressure Measurement in Smart Skin

**DOI:** 10.3390/s19081811

**Published:** 2019-04-16

**Authors:** Bao Yang, Su Liu, Xi Wang, Rong Yin, Ying Xiong, Xiaoming Tao

**Affiliations:** 1Research Centre of Smart Wearable Technology, Nanotechnology Center of Functional and Intelligent Textiles and Apparel, Institute of Textiles and Clothing, The Hong Kong Polytechnic University, Hong Kong, China; bao.yang@polyu.edu.hk (B.Y.); suliu.liu@connect.polyu.hk (S.L.); rryin@polyu.edu.hk (R.Y.); ying-xy.xiong@connect.polyu.hk (Y.X.); 2Engineering Research Center of Digitized Textile & Apparel Technology, Ministry of Education, College of Information Science and Technology, Donghua University, Shanghai 201620, China; xiwang@dhu.edu.cn

**Keywords:** pressure sensor, fibre Bragg grating, high sensitivity, high durability, soft matrix

## Abstract

Precise measurements of low pressure are highly necessary for many applications. This study developed novel structured fibre sensors embedded in silicone, forming smart skin with high sensitivity, high durability, and good immunity to crosstalk for precise measurement of pressure below 10 kPa. The transduction principle is that an applied pressure leads to bending and stretching of silicone and optical fibre over a purposely made groove and induces the axial strain in the gratings. The fabricated sensor showed high pressure sensitivity up to 26.8 pm/kPa and experienced over 1,000,000 cycles compression without obvious variation. A theoretical model of the sensor was presented and verified to have excellent agreement with experimental results. The prototype of smart leg mannequin and wrist pulse measurements indicated that such optical sensors can precisely measure low-pressure and can easily be integrated for smart skins for mapping low pressure on three-dimensional surfaces.

## 1. Introduction

Precise measurements of low pressure are highly necessary for many applications involving human interaction with industrial machines, vehicles in traffic accidents, robotic exoskeletons, contact sports, daily activities and healthcare applications [1,2,3,4,5,6]. The exerted pressure on a human body widely ranges from below one pascal to hundreds of kilo-pascals (kPa) or higher; for example, acoustic pressure in ears is normally below one pascal, and plantar pressure can be over hundreds of kPa when the people is running or jumping [7,8]. Excessive pressure, that is, pressure of about 4.33 kPa and above that induces capillary closure, will make people feel uncomfortable, have numbness of the affected body part, or even suffer from series health issues [9,10,11], and on the contrary, insufficient pressure will limit the efficacy of treatments. These differences highlight the significance of the precise measurement of pressure. Extensive research has been devoted to developing sensors, converting variation of the excreted pressure to perceptible signals including mechanical [12], electrical [3,12,13,14,15,16] and optical signals [2,5,17,18,19,20,21], for precise measurement. Relative to electrical sensors [14,15,16], optical fibre sensors have specific advantages of high sensitivity, robustness, good immunity to electromagnetic interference, the intrinsic safety without electricity at the measuring point, and ease of integration, that is, one optical fibre can have multiplexed strain/temperature sensing units by using the technologies of wavelength-division-multiplexing, fibre Bragg gratings and others. Such optical sensors have been demonstrated to monitor low pressure in the healthcare fields, such as measurements of planter pressure [2], blood pressure [22] and human vital signs [23], showing immense potential to form the sensing network for smart skins for precise measurement of low-pressure at human–machine, human–building, human–human and human–garment contact.

Various types of pressure sensors have been investigated based on fiber Barrage gratings (FBGs). The sensitivity of the FBG sensor, which is fabricated either by using FBGs alone or FBGs combined with a conventional transfer-structure, is too small for precise measurement of low-pressure. For example, the sensitivity using FBGs written on the conventional single mode fibre is only ~4 pm/MPa [24], and the sensitivity can be improved to ~13 pm/MPa using a six-hole suspended-core fibre [25], ~44 pm/kPa using a single-ring suspended fibre [26], and ~200 pm/MPa using polymer optical fiber [27]. Through an effective transfer-structure, we reported that a pressure sensor with lab-made polymer fiber Bragg gratings embedded in silicone, an ultrahigh pressure sensitivity was demonstrated up to 0.8 pm/Pa (800,000 pm/MPa) in the range of 0–2.4 kPa [17]. However, the size of such transfer-structure was large and thus the fabricated sensor was inconvenient for integration of smart skins. Furthermore, compared with commercial FBGs made of silica optical fibers, the lab-made polymer fiber Bragg gratings has a poor humidity stability, a short life and high cost. Therefore, it is highly desirable to develop an effective transfer-structure for small-size pressure sensors based on commercial silica FBGs.

Embedded in a soft matrix that forms a smart skin, the sensor acquires flexibility, softness and ease in integration. A smart skin normally requires numerous sensors for pressure mapping, in which one optical fibre integrates several sensors. The embedded sensor may behave drastically different from the stand-alone one, because of variation of constraint conditions and crosstalk. The physical properties of the sensing unit, such as elastic modulus, flexibility and thermal expansion coefficient, are normally different from those of matrix. These mismatching physical properties of optical fibre and soft matrix cause that challenges remain. For example, the elastic modulus of optical fibre made of silica is three orders higher than that of soft matrix (silicone). Stress/strain of optical fibre induced at adjacent zones of the sensing unit will transmit along the optical fibre without enough reduction in amplitude. This phenomenon drastically influences measurement. Furthermore, an effective transfer-structure that induces strain at gratings is necessary when low pressure is applied to a soft matrix. If temperature changes, the stress state among optical fibre, matrix and other components also varies due to different thermal expansion. Such variations cause that the responses of optical sensor drastically deviate from those of stand-alone sensors. Although previous studies reported the application of optical sensors [17,22,23,28,29], with high sensitivity, they mainly research behaviors of the stand-alone sensor. Therefore, a holistic design is highly necessary for developing a stable and sensitive sensor integrated in smart skin for the precise measurement of low-pressure.

This paper develops a novel structured fibre sensor embedded in silicone for the precise measurement of low pressure. The fabricated sensor has high sensitivity, good repeatability and good durability. A theoretical model of such optical sensor is presented and verified to have an excellent agreement with the experimental results, which can be a powerful tool to guide the structural design of devices and the selection of materials utilised in the structured fibre sensor for intended applications. The demonstrations of a smart leg mannequin and pulse wave measurements show the potential applications of smart skin for mapping low-pressure on three-dimensional surface.

The paper is organized as follows. The design and theoretical treatments of optical sensors are first introduced in Section 2. The calibration of the fabricated optical sensors as well as durability, temperature effect and mutual influence of optical sensors are discussed in Section 3. Two applications are demonstrated in Section 4. Finally, conclusions are drawn in Section 5.

## 2. Materials and Methods

### 2.1. Structural Design

To enhance the stability and sensitivity of structured fibre sensors, we introduced an effective transfer-structure, that converts the applied pressure into the axial tension of gratings. The proposed sensor, shown in Figure 1a, comprises an optical fibre with Bragg gratings, a rigid base with a rectangular groove, a spacer, a thin film on the base and matrix, where the thin film and matrix are made of same soft materials. This architecture has the following merits. (1) Unlike those of the previous designs [17,22,23,28,29], the sensing behavior of FBGs can be localized at a small zone, that is, the groove on the rigid base. The rigid base as the transfer-structure can effectively isolate the deformation of soft matrix and optical fibre due to its high rigidity and the strong constraints between the base and optical fibre, and between the base and the thin film. Thus, such transfer-structure can effectively reduce crosstalk among sensors; (2) High sensitivity to pressure can be easily achieved, because after fabrication, an air pocket forms in the groove and a spacer is mounted between the thin film and the optical fibre, making the FBGs is easily deformable and highly sensitive to the variation of applied pressure; (3) The measured range of pressure and the sensitivity can be adjusted by selecting matrix with different elastic modulus, or by changing geometric parameters of components, including the size of the groove and the spacer, and the thickness the thin film; (4) Reducing the temperature effect can be achieved by selecting materials with proper thermal expansion and high elastic modulus for the base.

### 2.2. Materials and Fabrications

SMF-28^®^ ultra-optical fibre (Corning Inc., New York, NY, USA), made of silica and polymer coating, is selected as optical fibre, due to its good mechanical and thermal reliability. The length of gratings is about 3 mm, and the Bragg wavelength is 1548 ± 0.5 nm. The thin film and matrix are made of silicone (Type of 903, Dongguan Xinrun Group Limited, Dongguan, China), because this material has an approximately linear relationship between stress and strain and low hysteresis, as shown in Appendix A. The base is made of Acrylonitrile Butadiene Styrene (ABS) or invar, whose elastic modulus is several orders higher than that of silicone. Photo paper is used as the spacer, due to its elastic modulus of dozens of megapascal in compression, in the medium between those of silicone and optical fibre, smooth, and ease of fabrication.

Fabrication of the proposed sensor includes seven major steps. Step 1: A silicone film measuring 50 mm in length, 20 mm in width and 1 mm in thickness was manufactured by using a Teflon mould. Step 2: Bases made of ABS or invar were manufactured by 3D numerically controlled machine tools. These bases have a rectangular groove with 20 mm in length, 5 mm in width and 1 mm in depth. Step 3: Optical fibre with Bragg gratings was straightened by stretching and then fixed on the base using instant glue (Aron Alpha, Toagosei Co., Ltd., Japan). The gratings were placed at the centre of the groove. It is noted that the coating layer of optical fibre at the parts connected to the base was stripped away, to ensure a good connection between optical fibre and the base. Step 4: A spacer was fixed on the centre of the silicone film, and then the silicone film was fixed on the top surface of the base using a small amount of silicone sealant. An air pocket was thus formed in the groove. Thereafter, the optical fibre in the groove achieved a slight pre-tension below 0.05% strain. Step 5: The fabricated unit was placed on the center of the mold, fixed by some silicone mixture (Type of 903, A: B = 100:1 in weight ratio). Additional silicone mixture was then added into the mould unit it was full. Step 6: The whole set was placed on a horizonal table at room temperature of 22 ± 1 °C for more than 8 h, until all the silicone was curing, meanwhile, and all air bubbles were removed. Step 7: The optical sensor was demoulded.

### 2.3. Theoretical Analysis

#### 2.3.1. Construction of Theoretical Models

The facture strain of an optical fibre made of silica is about 0.6% [30,31], showing the upper limit of the applied strain in measurements. Meanwhile, the durability of optical fibre sensors mainly relies on the level of the working strain [31]. The measured ranges in applications are normally different, such as the pressure applied by garments ranges from unit to several kPa [9,11], whereas the foot plantar of human can be up to 3 MPa [8]. Therefore, reasonable sensitivity is required in applications. And thus, it is highly desirable to construct a theoretical model to guide the design of the structured fibre sensor for intended applications.

To construct the theoretical model, several assumptions are made. First, all components only experience small deformation due to the small fracture strain of optical fibre made of silica. Accordingly, assuming that all components are made of linear-elastic materials is reasonable. Secondly, optical fibre is well fixed on the base by instant glue. Such fixed optical fibre with gratings in the groove can be considered as an elastic beam with the built-in condition at both ends. Thirdly, the film over the groove is simplified as a simply supported plate, because only part of bottom surface of the film is fixed on the base, whereas the top surface of the film is free. Fourthly, the groove has enough depth, thus, optical fibre does not touch the bottom of the groove during measurements. Fifthly, the influences of the spacer on the flexural stiffness on optical fibre and the film are neglected. Moreover, the spacer is sufficiently thick to separate optical fibre and the film over the groove. Sixthly, the wavelength shift of FBGs induced by radical pressure from the spacer can be neglected [18]. On the base of these assumptions, three cases are included in theoretical analysis: a simply supported rectangular plate with a load uniformly distributed over a rectangle zone, which corresponds to the size of the spacer (Figure 1b), a simply supported rectangular plate under a uniform pressure (Figure 1c), and a built-in beam under a uniform load over the centre part (Figure 1d).

From the theoretical treatment shown in Appendix A, the average stain, *ε*, of FBGs in the axial direction can be estimated by
(1)$$\varepsilon =\frac{1}{a}\left(\int_{-a/2}^{a/2}Wdx-a\right),$$
where *a* is the length of the groove, and *W* represents the deflection of optical fibre in the groove. *W* is a complex function of the applied pressure, *p*, the structural parameters and mechanical properties of the components of the optical sensor, thus, the possessing *ε* is equally complex. Then the wavelength shift, Δ*λ*, of FBGs, induced by *ε* and the change of temperature, Δ*T*, is given by the following relation [32,33].

(2)$$\Delta \lambda =\left(C_{s}\varepsilon +C_{T}\Delta T\right)\lambda_{B},$$
where *λ_B_* is the initial Bragg wavelength of FBGs, *C_S_* is the coefficient of strain, and *C_T_* is the coefficient of temperature change.

#### 2.3.2. Effects of Parameters in Simulation

Figure 2 presents the effects of parameters, including parameters of structures and mechanical properties of the components, on the wavelength shift of FBGs based on the above theoretical models. The parameters of the structure and materials utilised in calculation are listed in Table 1. As shown in Figure 2a, the contour lines of the wavelength shift of FBGs indicate that a larger value of the wavelength shift is normally obtained at a larger groove width, *b*. Moreover, the contour lines have valleys or monotonically increase with the increase in the groove length, *a*, indicating an optimal value of *a* for obtaining the highest value of the wavelength shift when *b* is given. For example, when *b* is 4.5 mm, the peak value of the wavelength shift of 500 pm is obtained at *a* of 7.5 mm, and when *b* is 3 mm, the highest value of 484 pm is obtained at *a* of 5 mm. The effective stiffness of the thin film, *D*, which is a combined factor of elastic modulus of *E*, Poisson’s ratio of *μ* and thickness of *h*, shown in Appendix A, plays an important role on the wavelength shift: with reducing the value of *D*, the wavelength shift has a drastic increase. As shown in Figure 2b, when *D* decreases from a unit to a half unit of *D*_0_, the wavelength shift increases from 200 to 555 pm at the applied pressure of 10 kPa. With the decrease in width, *a_s_*, and length, *b_s_*, of the spacer, respectively, the wavelength shift of FBGs increases monotonically and tends to be stable when the ratio of *b_s_/b* is below 0.2 in Figure 2c and the ratio of *a_s_/b* is below 1.2 in Figure 2d. Because the hypothesis of uniform interaction force between the film and the spacer (Figure 1b) and between the spacer and the optical fibre (Figure 1d) have been simulated. Another hypothesis in which the average deflection of the zone on the film and the optical fibre in contact with the spacer is provided in Appendix A. Therefore, the ratio of size between the spacer and the film over the groove affects the wavelength shift of FBGs. When such ratio declines, the contact conditions in simulation approach the hypothesized uniform pressure and uniform deflection, resulting in the contact zone changes from a surface close to a line or a point and the possessing wavelength shift trends of FBGs becomes stable. Finally, as shown in Figure 2d, a large slope of curves of the wavelength shift and the applied pressure is obtained at a large thickness of the spacer, *h_s_*, due to a high pre-tension of FBGs.

##### Simulate Effect

**Table 1 sensors-19-01811-t001:** Parameters utilized in calculations.

Symbol	Value	Symbol	Value
Length of the groove, *a*	20 mm	Elastic modulus of the core of optical fibre, *E_c1_*	70 GPa
Width of the thin groove, *b*	5 mm	Poison’s ratio of the core of optical fibre, *μ_c1_*	0.17
Elastic modulus of the thin film, *E*	1.5 MPa	Diameter of the cladding layer of optical fibre, *d_c2_*	125 μm
Poison’s ratio of the thin film, *μ*	0.45	Elastic modulus of the cladding layer of optical fibre, *E_c2_*	70 GPa
Thickness of the thin film, *h*	1 mm	Poison’s ratio of the cladding layer of optical fibre, *μ_c2_*	0.17
Length of the spacer, *a_s_*	4 mm	Diameter of the coating layer of optical fibre, *d_c3_*	242 μm
Width of the spacer, *b_s_*	3 mm	Elastic modulus of the coating layer of optical fibre, *E_c3_*	2.5 GPa
Thickness of the spacer, *h_s_*	0.3 mm	Poison’s ratio of the coating layer of optical fibre, *μ_c3_*	0.34
Diameter of the core of optical fibre, *d_c1_*	8.2 μm	Coefficient of strain, *C_S_*	0.78 [33]

## 3. Results and Discussions

### 3.1. Calibration of Optical Sensors

A calibration system, comprising by a compressed-air source, an air pressure governor, an air tank with a digital pressure meter and a digital temperature meter, an optical integrator and a computer, is shown in Figure 3a. The optical interrogator (si 155-Micron Optic, Micro Optics Inc., Atlanta, GA, USA) utilised in tests has a high sampling rate of 1000 Hz and wavelength accuracy of 1 pm and can thus provide a reliable and accurate measurement for dynamic tests. Meanwhile, a prediction based on the theoretical model in Section 2.3 was performed to verify the model. As shown in Figure 3b, the prediction of the wavelength shift curve in red shows an excellent agreement with the experimental results, indicating that theoretical models can be an effective tool to guide the structural design and selection of material of optical sensor for intended applications. The state of the first measured point at 0 kPa of the loading cycle is defined to be the initial state for the corresponding cycle. The measured and theoretical results show a non-linear relationship between the wavelength shift and the applied pressure. However, such relationship is close to be a linear one, where the coefficient of determination in the linear fitting is over 96%, showing that the sensitivity to the applied pressure is about 26.8 pm/kPa, beyond two orders higher than the other FBG sensors fabricated using FBGs alone or FBGs combined with a traditional transfer-structure [24,25,26,27]. As shown in Figure 3b, the wavelength shift of FBGs is below zero when the applied pressure goes back to 0 kPa, indicating some residue compression is applied on the FBGs. The reason is that the used silicone naturally is a viscoelastic material rather than an ideal elastic material; as expected, after compression, it cannot recover to the initial state quickly. The selected silicone has mechanical hysteresis about 2% full-scale output within the range of 25 kPa (Appendix A). Thus, the sensor fabricated by such silicone also has similar properties. When the applied pressure goes back to zero, the film made of such silicone normally has not recovered to the initial state, inducing a residue compression on FBGs. Such residue compression will fully disappear after resting for several minutes, that is, ~10 min in this case.

The sensor normally was rested for enough long time before the first loading, while the rested time among the follow-up cycles was short. Thus, the data in the first cycle are slightly different from those in follow-up cycles, as well as the data in follow-up cycles are consistent with each other. As shown in Figure 3b, the curves indicate a hysteresis of about 10% full-scale-output (~1 kPa) and the largest hysteresis occurs in the zone from 0 to 2 kPa. The hysteresis becomes below 5% full-scale-output (<0.5 kPa) if the data in the first cycle were excluded. Such hysteresis is reasonable because of soft polymer used. And this hysteresis is lower than other electrical sensors of ~9% (~1.3 kPa) [34], and < 30 kPa [35]. As the phenomenon of the residue compression and hysteresis increase systematic errors, several scenarios are proposed to reduce the systematic error. (1) Since the largest hysteresis occurs in the zone from 0 to 2 kPa, the measured pressure in the applications can avoid this zone; an example is the measurement of pressure in compression stockings, which normally ranges from 2.4 kPa to 6.5 kPa or higher; (2) Only the data during the loading process are utilised for evaluations, whereas the data during the loading process are well consistent with each other; (3) Before tests, pre-compression can be performed on the sensor to avoid the first cycle, whereas the hysteresis of the cyclic test is about 10% full-scale-output (~1 kPa) and becomes below 5% full-scale-output (<0.5 kPa) if the data in the first cycle are excluded.

Compressed air is a good candidate for supplying uniform pressure for calibrations, especially when the measured object has a curved surface. However, a long time is needed for the compressed air in the tank to reach balance. Hence, it is not convenient for cyclic tests. Moreover, the pressure applied by a flat plate can be assumed to be uniform when the top and bottom surfaces of the optical sensor are also flat, and the deformation is small. Therefore, in this study, the cyclic test was conducted on the fabricated sensor using a machine (Keyboard Life Tester ZX-A03, Shenzhen ZXD Testing Equipment Co., Ltd., Shenzhen, China), which can provide a continues sinusoidal motion and record the force synchronously by using an additional load cell [36]. After 1,000,000 cycles, the fabricated sensor still works well. During the first 10,000 cycle, the peaks of the wavelength shift shows a good repeatability. After 10,000 cycles, the peak wavelength shift of the fabricated sensor slightly increases from 323 to 363 pm (Figure 3c), possibly because the environmental is uncontrolled, where temperature increases from 21.9 ± 1 to 22.9 ± 1 °C and the relative humidity also changes from 45% ± 5% to 67% ± 5%. Small negative peaks induced by adhesion between the moving plate and the top surface of optical sensors are also observed, as shown in the measured curves in blue. After 100,000 cycles, the negative peaks of curves in red and black increase because of increase in the adhesive force possibly induced by the increased relative humidity. The behaviour of the optical sensor under compression, that is, the positive peak, is crucial for applications. The positive peaks of the wavelength shift do not show an obvious degradation or increase, indicating that the fabricated optical pressure sensor has excellent durability and repeatability.

### 3.2. Temperature Effect

The coefficient of temperature for bare FBGs comprises the thermal expansion coefficient and the thermos-optic coefficient [32,33]. The wavelength shift of the proposed optical sensor is sensitive to strain and change of temperature, as Equation (2), where the coefficient of temperature becomes increasingly complex, because besides variations in material properties, the internal stress state also changes due to different thermal expansion.

Two samples with different basses made of ABS and invar, respectively, were utilised to illustrate the temperature effect on the wavelength shift. Invar has a proper thermal expansion coefficient (0.8~1.6 × 10^−6^/°C) [37], close to that of silica (0.55 × 10^−6^/°C) [33], while ABS has a high coefficient (72 × 10^−6^/°C). These experiments were performed in an environmental chamber, shown in Figure 4a, where the exerted pressure on the optical sensor was applied by weights. As shown in Figure 4b,c, with an increase in temperature, the Bragg wavelength of FBGs increases. Meanwhile, the hysteresis of loading-unloading curves becomes large; for example, the hysteresis of the curves in blue is small, shown in Figure 4b, when temperature increases from 21 to 62 °C, hysteresis of the curve in red become huge. Compared with the Bragg wavelength shown in Figure 4b, variations of the Bragg wavelength induced by change in temperature are considerably small (Figure 4c), because the base plays an important role in the thermal properties of the optical sensor due to its high rigidity. Adoption of materials with proper thermal expansion coefficients, such as invar, can be an effective way to reduce the influence of thermal expansion. However, the hysteresis also increases with the rise in temperature, making that the correction for precise measurement is too complex. The measured deviation induced by the change of temperature still needs more investigations due to the complex thermal properties of martials.

### 3.3. Inflence of Tension on Fibre

A smart skin normally requires numerous sensors for pressure mapping, in which one optical fibre normally integrates several sensing units. The elastic modulus of optical fibre made of silica is about 70 GPa, several orders higher than that of silicone matrix (~1.3 MPa). Stress can be easily transmitted along the optical fibre from one sensor to the other without enough reduction, because of the weak constraint from soft matrix. Such mismatching physical properties of optical fibre and silicone matrix may induce crosstalk in an integrated sensing network. The crosstalk among optical sensors is demonstrated by stretching. The optical fibre was stretched at the end, as shown in Figure 5a. The load cell fixed on the machine (Intron 5566, Instron, New York, NY, USA) can move up and down to provide force pulses at the end of optical fibre (Figure 5b). The max force applied on the fibre is about 2.5 N, inducing a large strain of up to 0.25%. One sample without any base, that is, a bare fibre with FBGs embedded in the silicone matrix, and two samples with a frame base made of ABS and Invar, respectively, were utilised. The elastic modulus of Invar of ~110 GPa [37] is higher than those of silicone (~1.3 MPa), ABS (~3 GPa), and silica (~70 GPa). Figure 5c shows the peak wavelength shift decreases from 800 pm (without any base) to 90 pm (ABS base), even only 3 pm (invar base), indicating that the frame base can effectively block stress of the optical fibre. The reason is that the optical fibre is well fixed on the base. Hence, an attachment layer is formed between the optical fibre and the base. Such attachment layer is thin and has high elastic modulus. More importantly, the attachment area between the fibre and the base is much larger than the cross-sectional area of the fibre, making an effective constraint between the fibre and the base. Consequently, the base, the fibre and the attachment layer will be stretched together following the deformation compatibility condition. Therefore, the base plays an important role on the wavelength shift of FBGs under tension of optical fibre. The base has higher stiffness, the wavelength shift of FBGs shows smaller amplitudes.

## 4. Applications of Optical Pressure Sensors

Compression garments, exerting the required pressure on the surface of target body zones, have been researched and utilised in fields of healthcare, body-shaping, medical applications, and athletic applications [38]. Magnitude and durability of a pressure exerted on a human body are the key indicators for the compression garment and garment fit [39]. Insufficient pressure will limit efficacy whereas excessive pressure makes people feel uncomfortable, even causes damage to health [9]. The pressure applied from a compression garment greatly relies on the mechanical properties and in-location shapes of the human body or mannequin. Thus, small-sized and accurate pressure sensors are crucial for accurate measurement.

Apart from compact measurement systems, such as KIKUHIME, HOSY, HATRA and MST, which are based on pneumatic or electric sensors, novel measurement systems based on the proposed optical pressure sensor network (Figure 6a) can be a better choice to evaluate the actual pressure exerted by compression stocking, because of their high accuracy, bionics of the shape and mechanical properties, immunity to electromagnetic interference, and continuous monitoring. The pressure sensing network was designed according to the measurement points of the medical compression hosiery standard (Quality Assurance RAL-GZ 387/1). The pressure mapping was measured through two optical channels (front and back), each channel has four pressure sensors, as shown in Figure 6a. The results indicated that wavelength shift of FBGs has a linear relationship on the applied pressure. The structured fibre sensors embedded in the leg mannequin slightly differ in terms of the sensitivity of pressure, *C_p_*, ranging from 20 to 27 pm/kPa, with an accuracy of about 0.05 kPa on the measured pressure. Thus the exerted pressure, *p*, from compression stockings can be estimated easily form the wavelength shift of FBGs.
(3)$$p=\Delta \lambda /C_{p}.$$
when the compression stocking was placed on the leg mannequin, the exerted pressure will be evaluated continually through the optical integrator, and the evaluated result will be listed on a simple user interface for visible observation.

Fibre Bragg grating-based sensors have demonstrated their potential in human health monitoring, such as in ballistocardiographic measurements [40], blood-pressure evaluation [22], and blood glucose evaluation [41]. Pulse palpation is another potential application, which is an important part of vascular physical examination, which can offer substantial useful information [42], such as over 20 types of pulses identified in Chinese medicine. These diagnoses are all essentially based on the detected pulse wave signals, such as the arterial pulse waveform reflecting the systolic and diastolic blood pressure [22], and differential wavelength shift of the measured pulse wave singles, which can also be utilised to evaluate the blood glucose [41]. Thus, a precise measurement on the pulse wave signals will be a solid base to boost the accuracy of diagnosis through the same methods in the reference [22,41,42] and to develop new methods. However, vibrations induced by pulse waves normally are extremely weak. Without an effective transduction mechanism, the measured signals have low magnitude and are easily affected by environmental factors. The laboratory-fabricated FBG pressure sensor can be fixed on a wrist strap or be fixed simply on the wrist by elastic tape as shown in Figure 6b, where the peak of wavelength shift induced by the measured pulses is up to 50 pm, which is 15 times higher than the measured value using optical sensors [28,41]. It means that the proposed optical sensor highly sensitive to the vibration induced by the pulse waves, and thus provides an accurate measurement of the pulse waves, which is benefit of various applications, including diagnosis of heart rates, the pulse pressure [22], the blood glucose [41] and others [42].

## 5. Conclusions

In summary, we have developed novel optical sensors embedded in silicone, forming a smart skin with merits of high sensitivity, good durability and good immunity to crosstalk. A theoretical model for the structured fibre sensors was presented, showing an excellent agreement between the prediction and experimental results, which can effectively guide the design of such sensor in structures and selection of materials. The influences of temperature and stress from optical fibre were investigated, showing the use of a material with high elastic modulus and low thermal expansion for the frame base of the optical sensor is highly desirable. Prototypes of a smart leg mannequin and wrist pulse measurement show that such structured fibre sensor can accurately measure of the low pressure in these applications.

## Figures and Tables

**Figure 1 sensors-19-01811-f001:**
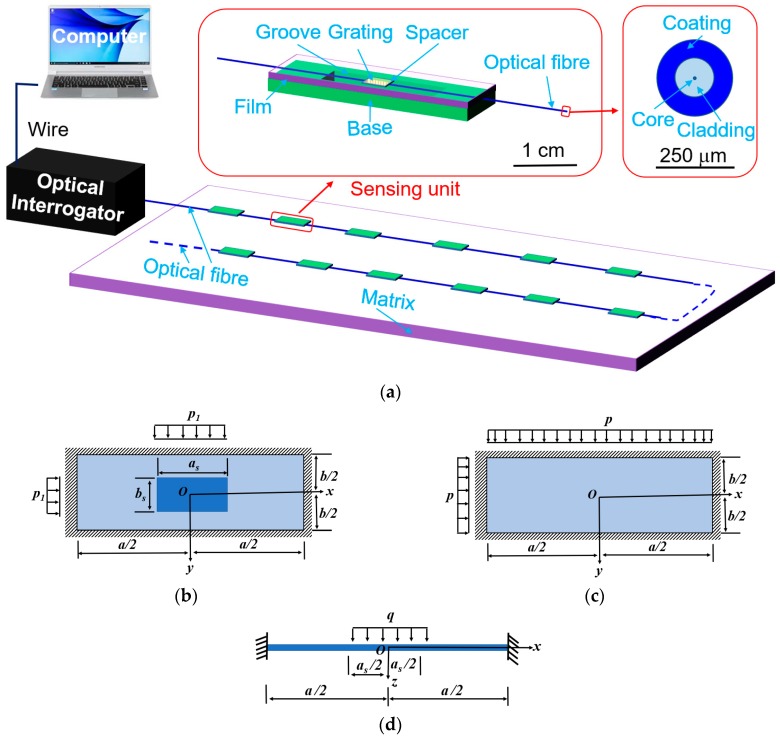
(**a**) Schematic of the smart skin with optical sensing networks and a measurement system, the proposed optical sensor, and the cross-section of optical fibre; Three models are utilised in theoretical analysis: (**b**) A simply supported rectangular plate under uniform load, *p*_1_, uniformly distributed on a rectangle zone at centre, which is corresponding to the size of the spacer; (**c**) A simply supported rectangular plate under uniform pressure; *p*, and (**d**) A built-in beam under a uniform load, *q*, at the centre part, which corresponds to the length of the spacer.

**Figure 2 sensors-19-01811-f002:**
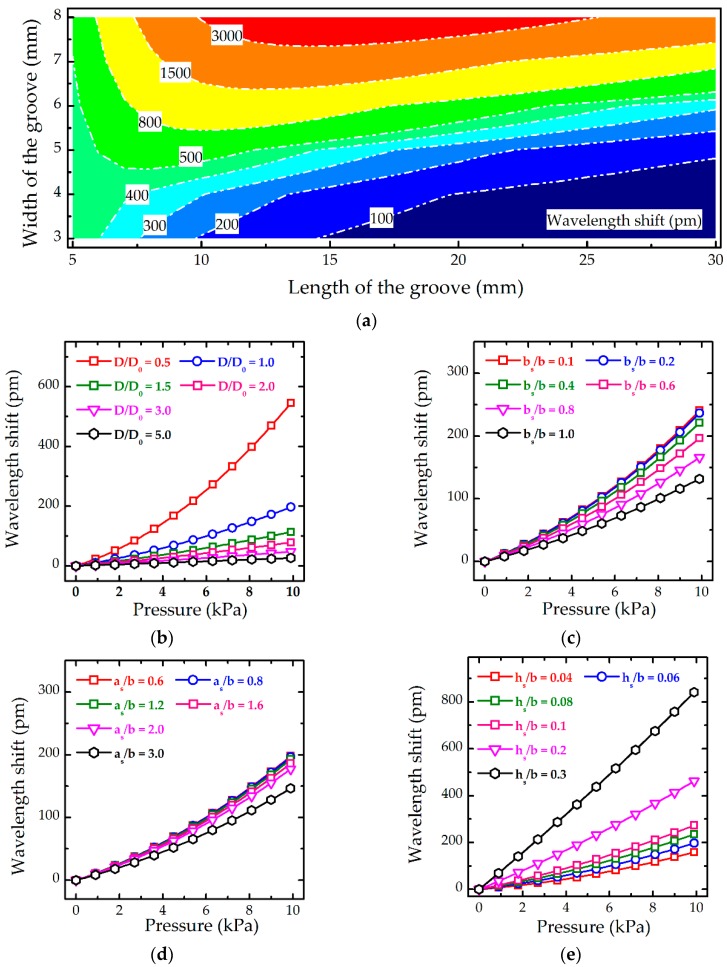
Effects of parameters on the wavelength shift of the proposed optical sensor based on the theoretical models: (**a**) length, *a*, and width, *b*, of the groove, where the wavelength shift of FBGs is obtained at the applied pressure of 10 kPa; (**b**) Effective stiffness of the thin plate, *D*, where *D*_0_ is calculated based on the parameters listed in Table 1; (**c**) Width of the spacer, *b_s_*; (**d**) Length of the spacer, *a_s_*; and (**e**) Thickness of the spacer, *h_s_*. Without specification, all the parameters utilised in calculation are listed in Table 1.

**Figure 3 sensors-19-01811-f003:**
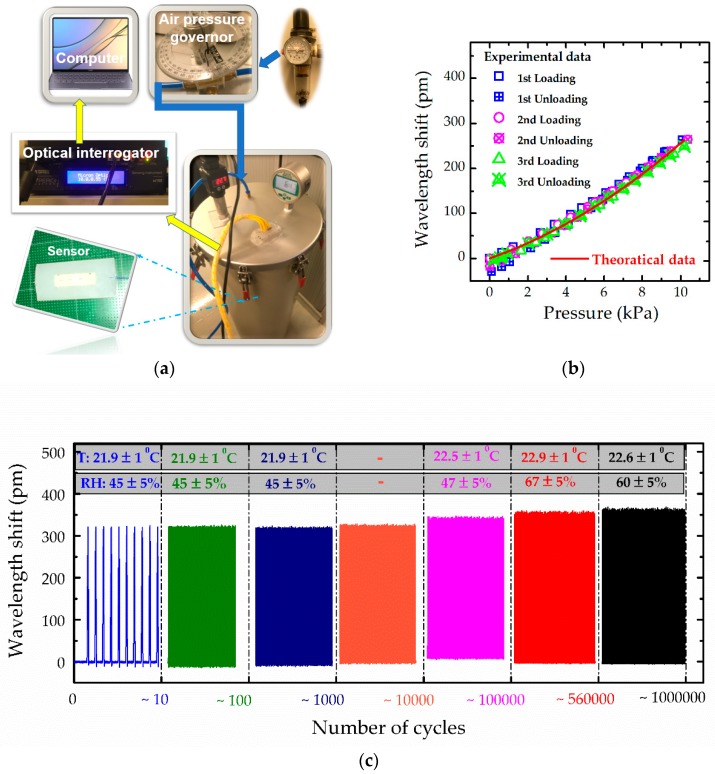
(**a**) Schematic of the calibration system; (**b**) Comparison between experimental results and theoretical results, where the thickness of the film is about 1.01 mm, and elastic modulus of the film is 1.3 MPa according to the experimental results shown in Appendix A, temperature is 22.6 ± 0.2 °C, and other parameters are consistent with those listed in Table 1; and (**c**) Results of cyclic compression, where the experiment was carried out on a keyboard life tester, setup in a room environment, the peak pressure is about 20 kPa and the frequency is 2.2 Hz.

**Figure 4 sensors-19-01811-f004:**
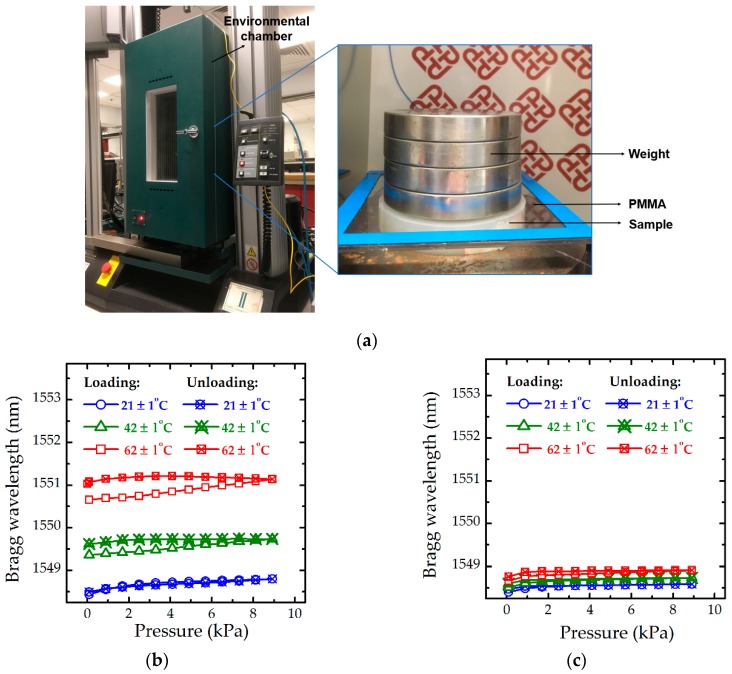
(**a**) Schematic diagram of experimental setup; (**b**) Effect of temperature of the optical sensor with an ABS base; and (**c**) Effect of temperature of the optical sensor with a base made of invar.

**Figure 5 sensors-19-01811-f005:**
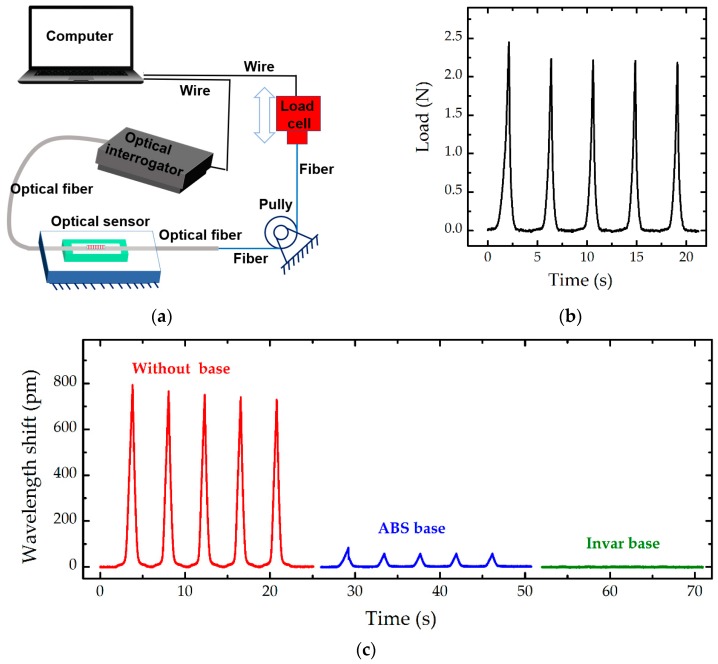
(**a**) Schematic of the designed loading system, aiming to analyze the influence of stress transmitted along the optical fibre on the output performance; (**b**) Illustration of the applied force at the end of optical fibre; and (**c**) Comparison among the measured results of three samples: one without any base (in red), the other two with a base made of ABS (in blue) and Invar (in green), respectively, at the room environment of 22.0 ± 1 °C and 65 ± 5% RH.

**Figure 6 sensors-19-01811-f006:**
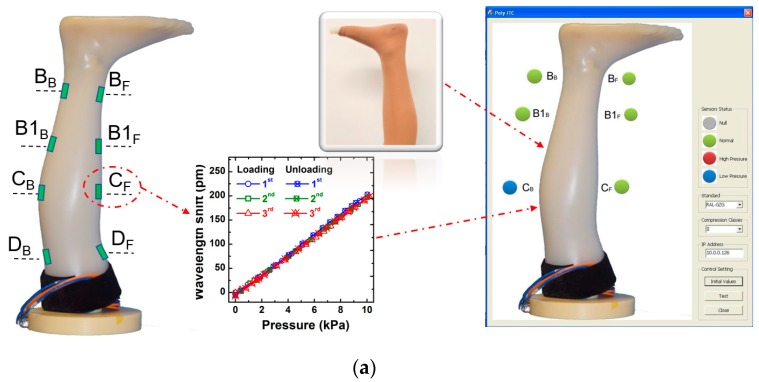
(**a**) Photo of a fabricated leg mannequin with optical sensors, which are setup based on the standard (Quality Assurance RAL-GZ 387/1), marked by B_F_~D_F_ and B_B_~D_B_ in the front and back side, respectively; Illustration of the calibration result of one fabricated optical sensor on the system (Figure 3a); Photo of compression stockings put on the fabricated leg mannequin, and a simple user interface for measurement of compression stockings, showing the pressure exerted on leg: green light is normal pressure, red light is high pressure in and blue light is low pressure; (**b**) Illustration of the application of wrist pulse measurement, where the thickness of the film over the frame base decreases to 0.4 mm for enhancing the sensitivity on pressure.

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
