# Peer review of "Highly Sensitive and Durable Structured Fibre Sensors for Low-Pressure Measurement in Smart Skin"

_sensors, 2019, doi:10.3390/s19081811_

Round 1
Reviewer 1 Report
In the work under revision the authors present an interesting study on the encapsulation parameters of silica optical fiber based FBG sensors, towards its application for lower pressure monitoring. As this is the re-submission of a previously reviewed paper, I consider that the authors addressed the comments made at the previous manuscript. However, still some moderate English revision should be made.
Also, the authors should explain why, in Figure 3b), the wavelength shift for the 0kPa (initial state) is <0. Such values are the indication of an initial compression on the FBG. The authors should comment on this observation and on how it may affect the sensors application in case scenarios as the ones presented in section 4.
Author Response
Detailed responses to the reviewer’s comments
We would like to thank referees for the careful review of our manuscript and providing us with constructive comments and suggestions to improve the quality of the manuscript. The following responses have been prepared to address the referees’ comment in a point-by-point fashion. And the changes in the manuscript have been highlighted with red colour.
Reviewer #1
In the work under revision the authors present an interesting study on the encapsulation parameters of silica optical fiber based FBG sensors, towards its application for lower pressure monitoring. As this is the re-submission of a previously reviewed paper, I consider that the authors addressed the comments made at the previous manuscript. However, still some moderate English revision should be made.
Reply: Thank the review for carefully reading and positive feedback. The authors have been carefully revised the manuscript. And the revised parts have been highlighted with red colour in the manuscript.
Also, the authors should explain why, in Figure 3b), the wavelength shift for the 0kPa (initial state) is <0. Such values are the indication of an initial compression on the FBG. The authors should comment on this observation and on how it may affect the sensors application in case scenarios as the ones presented in section 4.
Reply: We appreciate the reviewer’s careful reading of the manuscript. As shown in Figure 3b, during the unloading process, the wavelength shift of FBGs is blow 0 when the applied pressure goes back to 0 kPa. The reason is that soft polymer, that is, silicone was utilized in fabrication. The used silicone naturally is a viscoelastic material rather than an ideal elastic material; as expected, after compression, it cannot recover to the initial state quickly. Therefore, the sensor fabricated using such silicone also have similar properties. When the applied pressure goes back to zero kPa, the film made of such silicone cannot recover to the initial state simultaneously, inducing a residue compression on FBGs and showing a small hysteresis on the wavelength shift.
Such hysteresis of the fabricated sensor slightly increases the systematic error and complexity in the measurement. These effects can be decreased by choosing the following strategies. (1) Since the largest hysteresis occurs in the zone from zero to 2 kPa, the pressure measurement in the applications can avoid this zone; an example is the pressure measurement of compression stockings in Section 4, which normally ranges from 2.4 kPa to 6.5 kPa or higher (reference of the standard of Quality Assurance RAL-GZ 387/1). (2) Only the data during the loading process are utilised for evaluations, whereas the data during the loading process are well consistent with each other; (3) Before tests, pre-compression can be performed on the sensor to avoid the first cycle, whereas the hysteresis of the cyclic test is about 10 % full scale-output (~1 kPa) and becomes below 5% full scale-output (< 0.5 kPa) if the data in the first cycle are excluded.
The above discussions have been added in Section 3.1 (Line 225-249).

Reviewer 2 Report
General comments:
In this manuscript, the authors presents optical fiber sensor based in fiber Bragg grating (FBG) for measure low pressure. Several authors describe the use of FBG embedded in several materials, in this manuscript the authors present in silicone. Thus, this solution isn’t novel but one variation using silicone. Otherwise, the authors use a commercial system to interrogate the sensor.
What is the increment in the sensing field of this configuration? The authors not clarify this.
The authors refers that this solution presents a sensor for low pressure, having high sensitivity, low hysteresis, good repeatability and durability. The results presents for the authors don’t show these.
The figures 2, 3 and 5 are confuse. The authors not explain clearly each. Some graphics crate confusion.
The authors present also a supplementary file with some results for simulation and theoretical study. The authors find this information helpful? For understand some statement in the manuscript is necessary this information, thus the authors needs introduce in the manuscript.
I recommend the authors rewrite the manuscript and present results that prove this is a sensor for low pressure with high sensitivity, low hysteresis, good repeatability and durability. One applications and experimental measurement is important. Change also the title of manuscript.
I recommend reject this manuscript in the present form.
Author Response
Detailed responses to the reviewer’s comments
We would like to thank referees for the careful review of our manuscript and providing us with constructive comments and suggestions to improve the quality of the manuscript. The following responses have been prepared to address the referees’ comment in a point-by-point fashion. And the changes in the manuscript have been highlighted with red colour.
Reviewer #2
In this manuscript, the authors presents optical fiber sensor based in fiber Bragg grating (FBG) for measure low pressure. Several authors describe the use of FBG embedded in several materials, in this manuscript the authors present in silicone. Thus, this solution isn’t novel but one variation using silicone. Otherwise, the authors use a commercial system to interrogate the sensor. What is the increment in the sensing field of this configuration? The authors not clarify this. The authors refers that this solution presents a sensor for low pressure, having high sensitivity, low hysteresis, good repeatability and durability. The results presents for the authors don’t show these.
The figures 2, 3 and 5 are confuse. The authors not explain clearly each. Some graphics crate confusion.
The authors present also a supplementary file with some results for simulation and theoretical study. The authors find this information helpful? For understand some statement in the manuscript is necessary this information, thus the authors needs introduce in the manuscript.
I recommend the authors rewrite the manuscript and present results that prove this is a sensor for low pressure with high sensitivity, low hysteresis, good repeatability and durability. One applications and experimental measurement is important. Change also the title of manuscript.
I recommend reject this manuscript in the present form.
Reply: We appreciate the effort of the reviewer. However, we cannot entirely agree with the comments made because of the following reasons in point form:
1. Several authors describe the use of FBG embedded in several materials, in this manuscript the authors present in silicone. Thus, this solution isn’t novel but one variation using silicone.
Reply: 1. It is true some earlier work, including our earlier work, reported embedding FBGs in a material for various purposes. Our present work is not just embedding them simply in the elastomer. The novelty is to use an effective transfer-structure to enlarge the sensitivity for precise measurement of low-pressure. The fabricated sensor based on such transfer-structure has merits, including:
(1) High sensitivity. The sensitivity of the structured fibre sensors that fabricated using commercial silica FBGs is up to 26.8 pm/kPa (26800 pm/MPa). Such sensitivity is much higher than the sensor fabricated using FBGs alone or FBGs combined with a traditional transfer-structure, such as, ~ 4 pm/MPa [1], ~ 13 pm/MPa [2], ~ 44 pm/kPa [3], and ~ 200 pm/MPa [4], and is slightly lower than the reported highest sensitivity of a polymer fiber Bragg grating sensor ~ 0.8 pm/Pa (800000 pm/MPa) [5]. However, the size of the polymer fiber Bragg grating sensor is large and thus it is inconvenient for integration of smart skins. Moreover, relative to the commercial FBG made of silica, the polymer fiber Bragg gratings show a weak stability, a short life and high cost. Therefore, such sensitivity of the fabricated sensor is high enough, whereas silica have high elastic modulus and small fractural strain. To make a clearer statement, more discussions were added in Section 1 (Line 49-60) and Section 3.1 (Line 221-225).
(2) Easily tunable sensitivity. The measured range of pressure and the sensitivity can be adjusted by selecting matrix with different elastic modulus, or by changing geometric parameters of components, including the size of the groove and the spacer, and the thickness the thin film.
(3) Good immunity to crosstalk. The sensing behavior of FBGs can be localized at a small zone, that is, the groove on the rigid base. The rigid base can effectively isolate the deformation of soft matrix and optical fibre due to its high rigidity and the strong constraints between the base and optical fibre, and between the base and the thin film.
(4) Good durability and repeatability. As shown in Figure 3c, the fabricated sensor still works well after 1000000 cycles.
2. Otherwise, the authors use a commercial system to interrogate the sensor. What is the increment in the sensing field of this configuration? The authors not clarify this.
Reply: We used the commercial systems, as they were good enough for this study. Thus, no increment was presented in the manuscript.
3. The authors refer that this solution presents a sensor for low pressure, having high sensitivity, low hysteresis, good repeatability and durability. The results present for the authors don’t show these.
Reply: We would like to thank the reviewer for reading this manuscript. But we cannot agree this comment because of the following reasons:
Firstly, as shown in Figure 3b, the fabricated sensor shows a pressure sensitivity up to 26.8 pm/kPa, that is, 26800 pm/MPa, over two orders higher than other FBG sensors, either using FBGs alone or FBGs combined with a traditional transfer-structure. They have a sensitivity in ~ 4 pm/MPa [1], ~ 13 pm/MPa [2], ~ 44 pm/kPa [3], and ~ 200 pm/MPa [4]. In one of our earlier work, a soft sensor comprising a lab-made polymer FBG embedded in a silicone block without interfacial bonding, a super high pressure sensitivity of 0.8 pm/Pa was achieved [5]. This is the highest value reported so far with FBG as sensing element. Indeed, the sensitivity of the present sensor is lower than our previous polymer FBG sensors[5]. However, compared with the lab-made polymer fiber Bragg gratings, the commercial silica FBG shows ultrahigh stability, no moisture effect, long life and low cost with mass production.
Secondly, Figure 3b indicates a reasonable hysteresis of about 10 % full scale-output (~ 1 kPa), because soft polymer of silicone was utilized. Such hysteresis is lower than the commercial electrical pressure systems of 14 - 25% [6], and other electrical sensors of ~ 9 % (1.3 kPa) [7], and < 30 kPa [8]. As we cannot find an effective benchmark of the hysteresis in FBG pressure systems, we delete the statement of low-hysteresis.
Thirdly, as shown in Figure 3c, the fabricated sensor still works well after 1000000 cycles, showing the fabricated sensor is durable. Before 10000 cycles, the peaks of the wavelength shift show extremely good repeatability. After 100000 cycles, the peak wavelength shift is changed, because the environment of the measurement system is uncontrolled, whereas temperature slightly increases. This result indicates that the fabricated sensor can have a good repeatability.
Therefore, the demonstrated results in Figure 3 have shown the mentioned performance of high sensitivity, good repeatability and durability. To make a clearer statement, we revised the text in Section 3.1 (Line 219-268).
3. The figures 2, 3 and 5 are confuse. The authors not explain clearly each. Some graphics crate confusion.
Reply: According to the comments, we carefully checked and revised the description and discussion on the figure 2, 3 and 5. We would like to explain the construction of figure 2, 3, and 5 and their relationships below:
Figure 2 illustrates the simulation results by presenting the influences of the parameters, including parameters of structures and mechanical properties of the components, on the wavelength shift of FBGs. As the fiber made of silica is a commercial product, the parameters of optical fiber are constant in simulation. Besides, the base has been assumed to be rigid in Section 2.1(Line 92-95). And the influence of the spacer on the flexural stiffness on optical fiber and the film are neglected, shown in Section 2.3.1 (Line 157-158). The mechanical properties and structural parameter of the thin film can be integrated as the parameter of the effective stiffness, D, shown in Section 2.3.2 (Line 182-185). Thus, we only need to investigate the influence of structural parameters of the groove (Figure 2a) and the spacer (Figure 2c ~ e) and the effective stiffness of the film (Figure 2d).
Figure 3 illustrates the experimental results of one fabricated sensor, including the calibration systems (Figure 3a), the corresponding calibration results (Figure 3b) and results of cyclic compression (Figure 3c). To make a clearer statement, more description and discussion of this figure are revised and shown in Section 3.1 (Line 219-249).
Figure 2 and Figure 3 show the performance of a single fabricated sensor in simulation and in experiment, respectively. Figure 5 shows the influence of tension on fibre, which demonstrates the crosstalk in the sensing network embedded in a smart skin. Figure 5a shows schematic of the designed loading system. Figure 5b illustrates the applied load on optical fiber. Figure 5c shows the results of the fabricated sensors with or without bases as the transfer-structure, indicating the base plays an important role on the wavelength shift of FBGs. The description and discussion of this figure are revised and shown in Section 3.3 (Line 299-321).
4. The authors present also a supplementary file with some results for simulation and theoretical study. The authors find this information helpful? For understand some statement in the manuscript is necessary this information, thus the authors needs introduce in the manuscript.
Reply: The derivation process of the theoretical model is useful but long. We agree that this information showed in the manuscript will give more detail to the theoretical models. However, the theoretical model just is one of explorations in this manuscript, excessive length on the theoretical models is undesirable and decreases the readability of the manuscript. Therefore, avoiding excessive length of the manuscript, two important mathematical expressions of Equation (1) and (2) are illustrated in this manuscript, showing the main mechanism of the fabricated sensor, and the derivation of the theoretical models is presented in the supplementary.
4- I recommend the authors rewrite the manuscript and present results that prove this is a sensor for low pressure with high sensitivity, low hysteresis, good repeatability and durability. One application is important and real measurement is important. Change also the title of manuscript.
Reply: Thank the reviewer for the comments and suggestions. We cannot entirely agree with the comments made because the experimental results of the fabricated sensor (Figure 3b and 3c) have been indicated the performance of high sensitivity, good repeatability and durability. To highlight the novelty, the title is revised to “Highly Sensitive and Durable Structured Fibre Sensors for Low-pressure Measurement in Smart Skin”.
Reference
1. Xu, M.G.; Reekie, L.; Chow, Y.T.; Dakin, J.P. Optical in-Fiber Grating High-Pressure Sensor. Electron Lett 1993, 29, 398-399, doi:DOI 10.1049/el:19930267.
2. Wu, C.; Guan, B.O.; Wang, Z.; Feng, X.H. Characterization of Pressure Response of Bragg Gratings in Grapefruit Microstructured Fibers. Journal of Lightwave Technology 2010, 28, 1392-1397, doi:10.1109/Jlt.2010.2042277.
3. Htein, L.; Liu, Z.; Gunawardena, D.; Tam, H.-Y. Single-ring suspended fiber for Bragg grating based hydrostatic pressure sensing. Opt Express 2019, 27, 9655-9664, doi:10.1364/OE.27.009655.
4. Bhowmik, K.; Peng, G.D.; Luo, Y.; Ambikairajah, E.; Lovric, V.; Walsh, W.R.; Rajan, G. Experimental Study and Analysis of Hydrostatic Pressure Sensitivity of Polymer Fibre Bragg Gratings. Journal of Lightwave Technology 2015, 33, 2456-2462, doi:10.1109/Jlt.2014.2386346.
5. Zhang, Z.F.; Tao, X.M.; Zhang, H.P.; Zhu, B. Soft Fiber Optic Sensors for Precision Measurement of Shear Stress and Pressure. Ieee Sens J 2013, 13, 1478-1482.
6. Bader, D.L.; Bouten, C.; Colin, D.; Oomens, C.W. Pressure ulcer research: current and future perspectives; Springer Science & Business Media: 2005.
7. Shu, Y.; Li, C.; Wang, Z.; Mi, W.T.; Li, Y.X.; Ren, T.L. A Pressure Sensing System for Heart Rate Monitoring with Polymer-Based Pressure Sensors and an Anti-Interference Post Processing Circuit. Sensors-Basel 2015, 15, 3224-3235.
8. Ramuz, M.; Tee, B.C.K.; Tok, J.B.H.; Bao, Z.N. Transparent, Optical, Pressure-Sensitive Artificial Skin for Large-Area Stretchable Electronics. Adv Mater 2012, 24, 3223-3227, doi:10.1002/adma.201200523.

Round 2
Reviewer 2 Report
I agree with last modifications. The manuscript can be publih in presente form.
This manuscript is a resubmission of an earlier submission. The following is a list of the peer review reports and author responses from that submission.
Round 1
Reviewer 1 Report
The authors present an interesting study, both theoretical and experimental, on the encapsulation parameters of silica optical fiber based FBG sensors, towards its application for lower pressure monitoring.
The study is well structured, nevertheless, an extensive English revision should be made, regarding typos and grammar mistakes. Also, some other technical questions should be addressed:
1- In section 2.1, figure 1a), the thin film should be indicated in the figure. Also, the dimensions of each component could be added for an easy read (if it does not overload the figure).
2- In section 2.2, lines 105 and 106, a reference or results backing that statement should be added.
3- In section 2.3.2, line 162 and 163, please confirm this statement, as it is confusing. Maybe in line 163, it should be when a) is 9mm… what also does not seem correct in the figure.
4- Regarding Figures 2 c) and 2 d), it looks like there is a limit after which the wavelength shift stabilizes (figure 2c, b=0.2 and figure 2d) a=1.2). The authors should comment on such behaviour.
5- Section 3.1, where the authors say “integrator”, it should be replaced by interrogator, as that is the correct designation of the equipment referred. The same correction should be made in figure 3a).
6- The reviewer does not agree with the statement in line 186 to 188, as the theoretical curve appears to exhibit an exponential behaviour rather than a linear one. Some considerations regarding these difference in the behaviour should also be given.
7- In Figure 3b) from the represented data, it looks like the wavelength shift for the 0kPa of the 1st unload is <0, indicating a compression on the FBG. The authors should comment on this observation and on how it may affect the sensors application in use case scenarios as the ones presented in section 4.
8- In section 3.3, since the fiber is glued to the base (ABS/Invar), shouldn’t this “attachment” have a major relevance on the results obtained with the described loads, rather than the base material itself?
9- Lines 293 to 295 appear to be out of the context (maybe a forgotten text).
Reviewer 2 Report
General comments:
In this manuscript, the authors presents optical fiber sensor based in fiber Bragg grating (FBG) for measure low pressure. Several authors describe the use of FBG embedded in several materials, in this manuscript the authors present in silicone. Thus, this solution isn’t novel but one variation using silicone. The authors present one possible application, that can be interesting, but the manuscript write is confuse. The authors refers that this solution presents a sensor for low pressure, having high sensitivity, low hysteresis, good repeatability and durability. The results presents for the authors don’t show these.
The authors present also a supplementary file with some results for simulation and theoretical study. The authors find this information helpful? For understand some statement in the manuscript is necessary this information, thus the authors needs introduce in the manuscript.
I recommend the authors rewrite the manuscript and present results that prove this is a sensor for low pressure with high sensitivity, low hysteresis, good repeatability and durability. One applications is important and real measurement is important. Change also the title of manuscript.
I recommend reject this manuscript